# Patient-Reported Outcomes of Palbociclib Plus Exemestane with GnRH Agonist versus Capecitabine in Premenopausal Women with Hormone Receptor-Positive Metastatic Breast Cancer: A Prospective, Open-Label, Randomized Phase ll Trial (KCSG-BR 15-10)

**DOI:** 10.3390/cancers12113265

**Published:** 2020-11-05

**Authors:** Soohyeon Lee, Seock-Ah Im, Gun Min Kim, Kyung Hae Jung, Seok Yun Kang, In Hae Park, Jee Hyun Kim, Kyoung Eun Lee, Hee Kyung Ahn, Moon Hee Lee, Hee-Jun Kim, Han Jo Kim, Jong In Lee, Su-Jin Koh, Yeon Hee Park

**Affiliations:** 1Division of Oncology-Hematology, Department of Internal Medicine, Korea University College of Medicine, Korea University Anam Hospital, Seoul 02841, Korea; soohyeon_lee@korea.ac.kr; 2Department of Internal Medicine, Cancer Research Institute, College of Medicine, Seoul National University Hospital, Seoul National University, Seoul 03080, Korea; moisa@snu.ac.kr; 3Division of Medical Oncology, Department of Internal Medicine, College of Medicine, Yonsei University, Seoul 03722, Korea; GMKIM77@yuhs.ac; 4Department of Oncology, Asan Medical Center, College of Medicine, University of Ulsan, Seoul 05505, Korea; khjung@amc.seoul.kr; 5Department of Hematology-Oncology, School of Medicine, Ajou University, Suwon 16499, Korea; kangsy01@ajou.ac.kr; 6Center for Breast Cancer, National Cancer Center, Goyang 10408, Korea; parkih@korea.ac.kr; 7Department of Internal Medicine, Seoul National University College of Medicine, Seoul National University Bundang Hospital, Seongnam 13620, Korea; jhkimmd@snu.ac.kr; 8Department of Hematology and Oncology, Ewha Womans University Hospital, Seoul 07985, Korea; oncolee@ewha.ac.kr; 9Division of Medical Oncology, Department of Internal Medicine, Gachon University Gil Medical Center, Incheon 21563, Korea; hkahn@gilhospital.com; 10Department of Internal Medicine, Inha University School of Medicine, Incheon 22332, Korea; moonhlmd@inha.ac.kr; 11Department of Internal Medicine, Chung-Ang University College of Medicine, Seoul 06974, Korea; heejun@cau.ac.kr; 12Division of Hematology and Oncology, Department of Internal Medicine, Soonchunhyang University Hospital, Cheonan 31151, Korea; hzmd@schmc.ac.kr; 13Division of Hematology-Oncology, Department of Internal Medicine, Wonju Severance Christian Hospital, Yonsei University Wonju College of Medicine, Wonju 26426, Korea; oncohem@yonsei.ac.kr; 14Department of Hematology and Oncology, Ulsan University Hospital, Ulsan University College of Medicine, Ulsan 44033, Korea; sujinkoh@uuh.ulsan.kr; 15Division of Hematology-Oncology, Department of Medicine, Samsung Medical Center, Sungkyunkwan University School of Medicine, Seoul 06351, Korea

**Keywords:** palbociclib, capecitabine, breast neoplasm, premenopause, patient-reported outcome measures

## Abstract

**Simple Summary:**

We reported that palbociclib plus exemestane with ovarian function suppression (with leuprolide) led to significantly longer median progression-free survival compared with capecitabine in premenopausal metastatic breast cancer patients. We also evaluated differences of patient report outcomes (PROs) between palbociclib plus endocrine therapy (ET) and capecitabine as upfront therapy in this study population. All the European Organization for Research and Treatment of Cancer quality of life questionnaire (EORTC QLQ-C30) were maintained from baseline to the end of treatment within each treatment arm. Patients with palbociclib plus ET arm experienced delay in time to deterioration of physical functioning, nausea and vomiting, and diarrhea. There was a trend for worsening of insomnia in the palbociclib plus endocrine therapy (ET) arm and appetite loss in the capecitabine arm. Patients with palbociclib plus ET had significant overall improvement of quality of life and significant delay in time to deterioration without compromising treatment efficacy.

**Abstract:**

In the era of CDK4/6 inhibitors in hormone receptor (HR)-positive, HER2-negative metastatic breast cancer, few trials have been specifically studied to compare quality of life between palbociclib plus endocrine therapy (ET) and cytotoxic chemotherapy exclusively in premenopausal women. We aimed to evaluate differences of patient report outcomes (PROs) between palbociclib plus ET and capecitabine. PROs were assessed using EORTC QLQ-C30 at baseline, every 6 weeks, and the end of treatment. All EORTC QLQ-30 scores were maintained from baseline to the end of treatment. Patients treated palbociclib plus ET arm experienced delay in time-to-deterioration of physical functioning (HR = 0.58, 95% CI, 0.36 to 0.84, *p* = 0.0058), nausea and vomiting (HR = 0.48; 95% CI, 0.32 to 0.73, *p* = 0.0005), and diarrhea (HR = 0.42; 95% CI, 0.27 to 0.65, *p* = 0.001). There was a numeric trend for worsening of insomnia (HR = 1.43; 95% CI, 0.96 to 2.16, *p* = 0.079) and favoring of appetite loss (HR = 0.69, 95% CI, 0.44 to 1.07, *p* = 0.09) in the palbociclib plus ET arm. Premenopausal patients with palbociclib plus ET maintained QoL without compromising treatment efficacy.

## 1. Introduction

Although breast cancer is known to be more aggressive and to be associated with a poorer prognosis in premenopausal women than in postmenopausal women, endocrine treatment (ET) is recommended as a standard of treatment by clinical guidelines for hormone receptor (HR)-positive, HER2-negative metastatic breast cancer (MBC) in premenopausal patients [1]. Nevertheless, 30–65% of premenopausal patients with HR+, HER2− MBC has been still treated with upfront chemotherapy across USA [2], European countries [3,4,5], and Korea [6] in the real world, even in the absence of visceral crisis. The inconsistencies between real-world treatment patterns and guideline recommendations may be partly due to a lack of direct comparisons of ET with chemotherapy and lack of clinical trials focused on premenopausal patients with breast cancer.

The emergence of modern cyclin-dependent kinase (CDK) inhibitors has changed the treatment paradigm for HR+ breast cancer. For premenopausal HR+, HER2− MBC patients, CDK4/6 inhibitors plus ET also have demonstrated superior progression-free survival (PFS) versus ET alone in pivotal trials such as MONALEESA-7 [7] and in subpopulations of PALOMA-3 [8] and MONARCH-2 [9]. However, premenopausal patients are underrepresented in those global trials such as 21% in PALOMA-3 and 16% in MONARCH-2. That is the reason why premenopausal HR+, HER2− MBC patients had limited options to choose the optimal therapeutic strategies under the lack of scientific evidence [10].

The YoungPearl trial was the first prospective randomized trial to directly compare treatment with palbociclib plus ET and capecitabine in premenopausal women with HR+, HER2− MBC. Palbociclib plus ET significantly prolonged the primary endpoint of progression-free survival (PFS) compared with that for capecitabine (median 20.1 months vs. 14.4 months, hazard ratio 0.659; 95% CI 0.437–0.994; one-sided log-rank *p* = 0.0235) [11].

Both ET and chemotherapy have the potential to negatively impact patient quality of life independent of clinical efficacy [12]. ET has provided muscle and joint pain and menopausal symptoms that may facilitate deterioration of quality of life (QoL) which might further negatively impact adherence, leading to early treatment discontinuation. Chemotherapy also worsens QoL and various toxicities such as nausea and vomiting, neuropathy, bone marrow suppression demonstrated through active treatment period and needed time to recover after chemotherapy discontinuation. Therefore, patient-reported outcomes (PROs) reflect patients’ perspective of their symptoms, function, and provide important complementary data to efficacy and safety endpoints.

We aimed to evaluate the patient reported outcomes (PROs) in the phase 2 trial, YoungPearl, particularly detailing the effects of palbociclib plus ET versus capecitabine on patients’ symptoms, functions, and global health status which could offer valuable insight into the therapeutic benefit of each regimen by measuring whether QoL is maintained during treatment.

## 2. Materials and Methods

### 2.1. Study Design and Participants

A detailed study design has been previously reported [11]. YoungPearl was a multicenter, randomized, open-label, phase 2 trial done in 14 academic institutions in South Korea. Premenopausal women aged 19 years or older with HR+, HER2− breast cancer that had relapsed or progressed during previous tamoxifen therapy with an Eastern Cooperative Oncology Group performance status of 0–2. One line of previous chemotherapy for MBC was allowed. Among 189 enrolled premenopausal MBC patients, 184 eligible patients were randomly assigned (1:1) to either palbociclib plus ET (*n* = 92) or capecitabine (*n* = 92). Six patients in the capecitabine arm withdrew from the study before drug administration; therefore, 92 patients in the palbociclib plus ET arm and 86 patients in the capecitabine arm were included in the modified intention-to-treat analyses. The primary objective was to demonstrate the superiority of palbociclib plus ET over capecitabine in prolonged investigator-assessed PFS in premenopausal HR+, HER2− MBC. The study was conducted in accordance with the Declaration of Helsinki and Good Clinical Practice guidelines and was approved by Korea Cancer Study Group institutional review board (KCSG BR-15-10) and institutional review board at each center. Trial Registration ClinicalTrials.gov identifier: NCT02592746.

### 2.2. Randomization and Study Treatments

Randomization was stratified by previous chemotherapy for metastatic breast cancer (yes versus no) and visceral metastasis (yes versus no). Patients were randomly assigned, using a random permuted block design (with a block size of two), to receive palbociclib plus combination ET (oral exemestane 25 mg per day for 28 days and oral palbociclib 125 mg per day for 21 days every 4 weeks plus GnRH agonist 3·75 mg subcutaneously every 4 weeks) or chemotherapy (oral capecitabine 1250 mg/m² twice daily for 2 weeks every 3 weeks).

### 2.3. Patient Reported Outcomes Assessment

PROs were assessed using the European Organization for Research and Treatment of Cancer Quality of Life Questionnaire Core 30 (EORTC QLQ-C30, version 3.0) [13] at baseline (≤7 days before cycle 1 day 1), every 6 week, and the end of treatment.

EORTC QLQ-C30 is a 30-item questionnaire comprising a global health status (GHS)/QoL scale (primary variable of interest), five multi-item functional subscales (physical, emotional, social, cognitive, and role), three multi-item symptom scales (fatigue, nausea and vomiting, and pain), and six single-item symptom scales assessing other cancer-related symptoms (dyspnea, insomnia, appetite loss, constipation, diarrhea, and financial difficulties). The questionnaire includes four-point Likert scales, with responses from “not at all” to “very much” to assess functioning and symptoms and two seven-point Likert scales for GHS/QoL. Responses to all items are converted to a 0–100 scale using a standard scoring algorithm. For functioning and GHS/QoL scales, higher scores represent a better level of functioning and QoL (a negative change from baseline reflects deterioration, and a positive change reflects improvement). For symptom scales, a higher score represents higher symptom severity (a negative change reflects improvement, and a positive change reflects deterioration).

### 2.4. Hormone Measurement

Estradiol and Follicle-stimulating hormone (FSH) were measured every cycle to check the menopausal status of the patients.

### 2.5. Statistical Analysis

According to the intent-to-treat principle, patients were analyzed according the treatment they were assigned to during randomization. Patients with an evaluable baseline score and at least one evaluable postbaseline score during the treatment period were included in the change from the baseline analyses, assessed by linear mixed effects models for repeated measure. Clinically meaningful change was defined as a ten-point or greater change from baseline in GHS/QoL, functioning, and symptom score. The TTD on GHS/QoL and functional scales was determined using Kaplan–Meier survival analysis methods from baseline to the first occurrence of a ten-point or greater decrease in the functional score, disease progression or death. TTD in symptom scales was determined using Kaplan–Meier survival analysis methods from baseline to the first occurrence of a ten-point or greater increase in the symptom score, disease progression or death [14]. The TTD, including median TTD with a two-sided 95% CI, was compared between the treatment arms’ survival distribution using the Kaplan–Meier method. A stratified Cox regression was used to determine the hazard ratio, with a two-sided 95% CI. Unless otherwise specified, analyses were conducted based on the observed data without imputation of missing data.

## 3. Results

### 3.1. Baseline Patients and Disease Characteristics

A total of 178 patients (92 in the palbociclib plus ET arm and 86 in the capecitabine arm) were included in the final PROs analysis. Baseline characteristics were well balanced across treatment arms (see Appendix A). Median age was 44 (28–58) years. Ninety-one (50%) of 178 received no previous treatment for MBC and 153 (86%) patients relapsed while on tamoxifen or within 12 months after completion of adjuvant tamoxifen, 88 (49%) patients had visceral metastasis (also see Appendix A).

From baseline to 84 weeks (21 cycles in palbociclib plus ET arm and 28 cycles in capecitabine arm), 100% of patients in the palbociclib plus ET arm and 94.2% in the capecitabine arm respectively, completed ≥1 EORTC QLQ-C30 assessment. Baseline scores of every function and symptom subscale scores as well as GHS/QoL were within range of reference values published previously in MBC patients (Table 1). There was no significant interaction effect between treatment and cycle, which suggests that the slope of the cycle was not different between treatment groups (see Appendix A).

### 3.2. Global Health Status/Quality of Life

The baseline scores of GHS/QoL were a little bit different in both treatment arms. Baseline GHS/QoL scores (standard deviation, SD) in palbociclib plus ET arm were higher 65.2 (20.8) than that of capecitabine 57.0 (22.3) with statistically significance (*p* = 0.0223 by Wilcoxon rank sum test) (Table 1). Over time, there was a short period of increasing trend at the beginning phase of the treatment but back to the decreasing trend in the remaining treatment period which was a similar pattern across both treatment arms (Figure 1). Through week 84, adjusted mean changes from baseline were within 5 points for all visits in both treatment arms. According to the analysis using generalized estimating equation (GEE), GHS/QoL was maintained from baseline to the end of treatment across all time points within each arm (*p* = 0.6230) (Appendix A). Results from the TTD analyses of GHS/QoL were also consistent with the observed mean changes and did not demonstrate statistically significance between the two treatment arms even though TTD of the capecitabine arm tended to be delayed (HR = 1.21, 95% CI, 0.80 to 1.86) (not demonstrated—KM survival graph).

### 3.3. Functioning Scales

Baseline functioning scale scores were high in both treatment arms and were generally consistent with the reference values (Table 1). In both arms, physical, emotional, and cognitive functioning scores in EORTC QLQ-C30, functioning scales were similar, whereas the role and social functioning were favored in palbociclib plus ET arm. Based on repeated measures mixed-effect model, there was no statistically significant change pattern according to time of all 5 functioning subscale scores (Figure 2); the physical functioning of the capecitabine showed a favored trend in terms of overall change from the baseline (Figure 3a). However, patients with palbociclib plus ET experienced delay in TTD in physical functioning with statistically significance (HR = 0.58, 95% CI, 0.36 to 0.84) which meant the physical function of palbociclib plus ET was well maintained without deterioration over the course of treatment (Figure 4a). According to TTD analyses of the other functioning scales, there was no statistical significance in the TTD of other functioning subscale scores between arms except physical functioning.

### 3.4. Symptom Scales

Mean baseline scores for symptom scales were similar except pain and generally low in both treatment arms (Table 1). In palbociclib plus ET arm, nausea and vomiting and diarrhea tended to be favorable in change from baseline scores. Fatigue, pain, dyspnea, and appetite loss were observed without statistically significant difference between-treatment difference (Figure 2); however, there was a trend for improving of insomnia and constipation in capecitabine arms (Figure 3b). Consistently, treatment with palbociclib plus ET significantly delayed TTD in nausea and vomiting (HR = 0.48; 95% CI, 0.32 to 0.73, *p* = 0.0005) and diarrhea (HR = 0.42; 95% CI, 0.27 to 0.65, *p* = 0.001) (Figure 4b). Other symptom scores indicated no statistical differences between treatment arms; there was a trend for worsening of insomnia in the palbociclib plus endocrine therapy arm (HR = 1.43; 95% CI, 0.96 to 2.16, *p* = 0.079) and the appetite loss tended to worsen in the capecitabine arm (HR = 0.69, 95% CI, 0.44 to 1.07, *p* = 0.09) (Figure 4c–e). The worsening of insomnia could be dependent on estradiol inhibition. Although both arms kept low FSH value as premenopausal level during the treatment period (mean FSH as 6.6 mIU/mL for palbociclib plus ET and 25.0 mIU/mL for capecitabine), palbociclib plus ET arm showed more potent than capecitabine in terms of estradiol suppression (mean estradiol as 23.9 pg/mL for palbociclib plus ET and 91.1pg/mL for capecirabine) which could be related with sleep disturbance (Figure 5).

## 4. Discussion

MBC, although treatable, is ultimately incurable so it is crucial to maintain QoL when new treatments emerge. The European Society for Medical Oncology Magnitude of Clinical Benefit Scale guidance emphasized the importance of a holistic assessment of the value of medicine that includes PROs in addition to efficacy and safety [15].

The combination of palbociclib plus ET significantly improved median PFS and the overall response rate compared with capecitabine in premenopausal HR+, HER2− MBC patients [11]. The PROs data presented here demonstrate that GHS/QoL as a surrogate of overall QoL was maintained in patients treated with palbociclib plus ET versus capecitabine without meaningful difference. Even though capecitabine as a cytotoxic drug reported a shorter PFS than palbociclib plus ET, the response has been maintained for a long time with well management of AEs in certain patients.

The differential impact of ET versus chemotherapy on QoL has not been fully characterized using validated tools to measure PROs. In this study, some difference profiles were observed between palbociclib plus ET and capecitabine in the EORTC QLQ-C30 functioning and symptom subscale scores. The TTD of patients with palbociclib plus ET had significant overall improvements and significant delay in many functioning and symptom scores especially in physical functioning (HR = 0.58, 95% CI, 0.36 to 0.84) and nausea and vomiting (HR = 0.48; 95% CI, 0.32 to 0.73, *p* = 0.0005) and diarrhea (HR = 0.42; 95% CI, 0.27 to 0.65, *p* = 0.001) with statistically significance. In other words, patients treated with capecitabine deteriorated in nausea and vomiting and diarrhea, which were consistent adverse event profiles in several pivotal trials [16,17]. The premenopausal HR+, HER2− MBC patients who received palbociclib plus ET could control their symptoms and maintained physical functions in their daily lives. Interestingly, there was a trend for worsening of insomnia in the palbociclib plus ET arm (HR = 1.43; 95% CI, 0.96 to 2.16, *p* = 0.079). Premenopausal patients who received palbociclib plus ET were more sensitive to menopause symptom caused by estrogen suppression which could be related with the worsening of insomnia rather than other symptom subscales. In contrast, patients treated with capecitabine could maintain a higher QoL in certain area with high levels of estradiol.

As is well known in the treatment of early breast cancer, chemotherapy led to cumulative, yet transient, QoL deterioration, which resolves shortly after treatment completion. In contrast with chemotherapy, ET had a more prolonged negative effect on QoL, finally leading to early treatment discontinuation [18]. Disease progression may negatively affect QoL and delaying progression could delay QoL deterioration in the metastatic setting. The addition of palbociclib to ET maintained good QOL in treatment-naïve or endocrine resistant patients with HR+, HER2− MBC compared with ET alone and significantly delay in health-related QOL was observed in patients without progression versus those who progressed [19,20].

These results have several strengths and limitations for interpretation. To our knowledge, this study is the first PRO data to directly compare palbociclib plus ET and chemotherapy in the frontline setting. Our data support treatment guideline recommendations as an active treatment option for premenopausal HR+ HER2− MBC with maintaining QoL, even though the EORTC QLQ-BR23 questionnaire was not included to identify breast cancer specific functions and symptoms. Second, this study demonstrated that ET with estrogen suppression could be tolerable without compromising treatment efficacy in premenopausal HR+, HER2− MBC patients with additional palbociclib incorporation.

Being an open-label trial, the PROs might be subject to patient biases. Patients’ knowledge of treatment received could influence their view and reporting of their symptoms. In addition, PROs data were only collected until progression, and time to disease progression was different between treatment arms. Therefore, the amount of data collected differs between the two treatment arms, potentially favoring the capecitabine arm with the shorter PFS, given the potential for a detrimental effect on PROs with longer exposure to palbociclib plus ET.

Despite the long treatment course, health-related QoL using EORTC QLO-C30 was maintained in patients treated with palbociclib plus ET. This study also showed better symptom subscales and physical functioning favoring palbociclib plus ET arm in TTD analysis. These results, combined with superior efficacy, indicate that palbociclib combined with ET is a desirable treatment option for premenopausal patients with HR+, HER2− MBC in the frontline setting.

## 5. Conclusions

We evaluated differences patient report outcomes (PROs) between palbociclib plus ET and capecitabine as upfront therapy in this study population. All EORTC QLQ-C30 were maintained from baseline to the end of treatment within each treatment arm. Patients with palbociclib plus ET arm experienced delay in time-to-deterioration of physical functioning, nausea and vomiting, and diarrhea. There was a trend for worsening of insomnia in the palbociclib plus ET arm and appetite loss in the capecitabine arm. Patients with palbociclib plus ET had significant overall improvement of QoL and significant delay in TTD without compromising treatment efficacy.

## Figures and Tables

**Figure 1 cancers-12-03265-f001:**
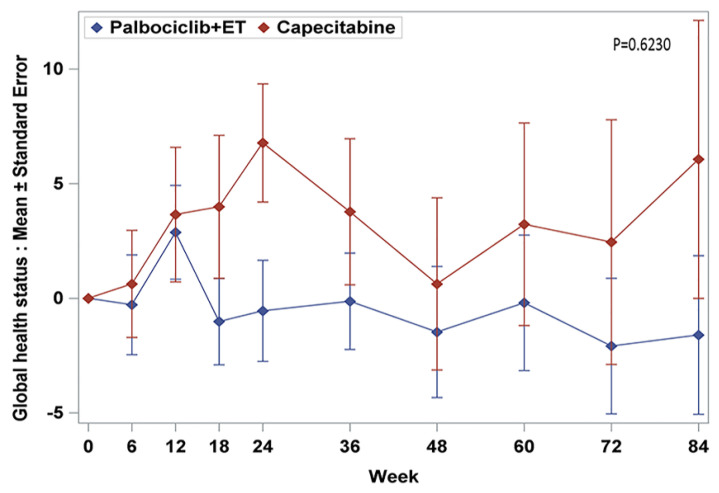
Change from baseline in EORTC QLQ-C30 global health status/quality of life.

**Figure 2 cancers-12-03265-f002:**
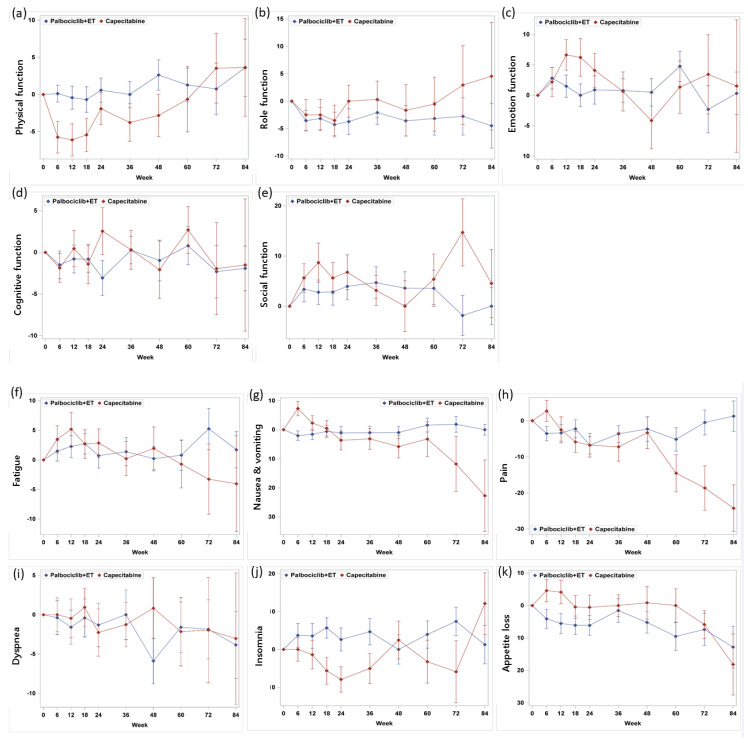
Change from baseline in EORTC QLQ-C30 function scales: (**a**) physical, (**b**) role, (**c**) emotion, (**d**) cognitive, (**e**) social function, symptom scales: (**f**) fatigue, (**g**) nausea and vomiting, (**h**) pain, and symptome scales: (**i**) dyspnea, (**j**) insomnia, (**k**) appetite loss, (**l**) constipation, and (**m**) diarrhea.

**Figure 3 cancers-12-03265-f003:**
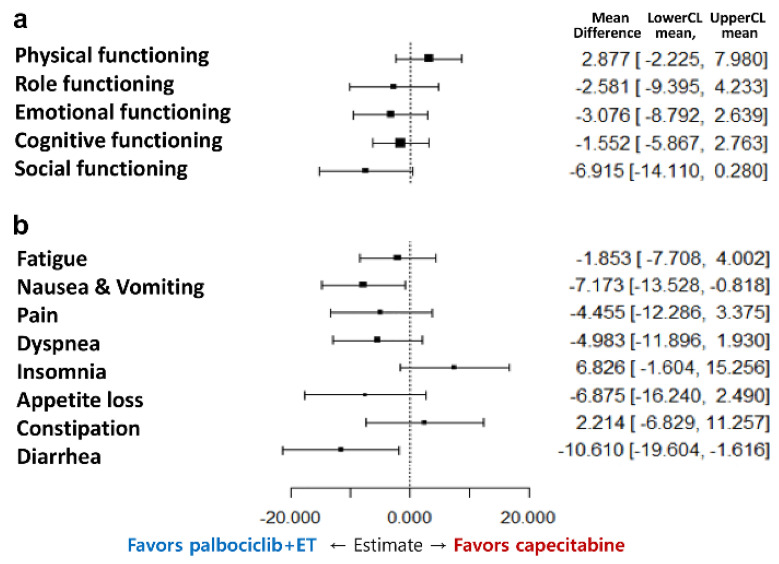
Forest plot model of estimated difference (palbociclib plus exemestane with GnRH agonist and capecitabine) in overall change from baseline in EORTC QLQ-C30. (Repeated-measure mixed-effect model) (**a**) functional and (**b**) symptom scales.

**Figure 4 cancers-12-03265-f004:**
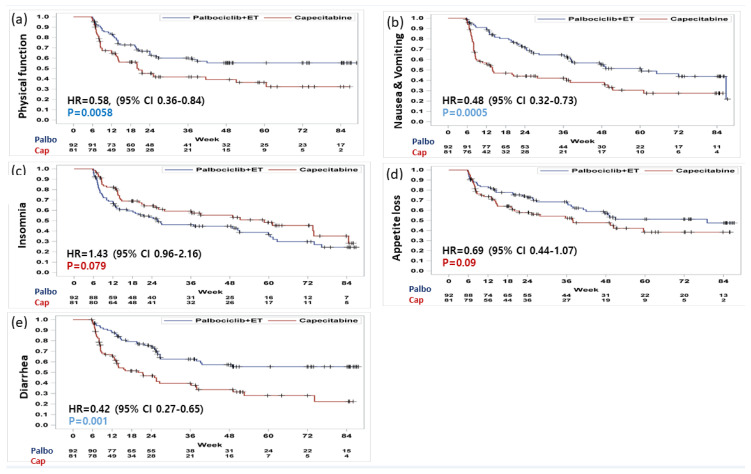
Time to definitive clinically meaningful deterioration in EORTC QLQ C-30. Deterioration in (**a**) physical function, (**b**) nausea and vomiting, (**c**) insomnia, (**d**) appetite loss, and (**e**) diarrhea.

**Figure 5 cancers-12-03265-f005:**
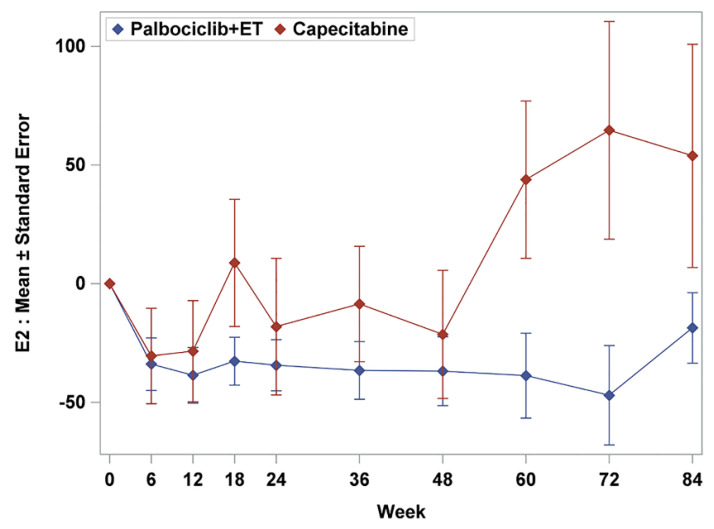
Estradiol change during the treatment.

**Table 1 cancers-12-03265-t001:** Baseline EORTC QLQ-C30 scores and reference values (PRO analysis set ^a^).

Domain/Scale	Palbociclib + Exemestane + GnRH Agonist (*N* = 92) Mean (SD)	Capecitabine (*N* = 86) Mean (SD)	*p*-Value	Reference Values ^b^ Mean (SD)
EORTC QLQ-C30 global health status/QoL ^c^				
Global health status/QoL	65.2 (20.8)	57.0 (22.3)	0.0223	60.2 (25.5)
EORTC QLQ-C30 functional scales ^c^				
Physical functioning	79.7 (18.3)	78.4 (20.0)	0.8244	81.6 (18.7)
Role functioning	82.3 (22.4)	74.3 (27.1)	0.0411	67.4 (31.1)
Emotional functioning	74.8 (18.7)	70.2 (21.6)	0.1266	65.9 (24.6)
Cognitive functioning	83.9 (15.7)	81.7 (18.0)	0.5060	80.5 (23.2)
Social functioning	77.0 (23.6)	66.3 (28.1)	0.0085	74.2 (28.4)
EORTC QLQ-C30 symptom scales ^d^				
Fatigue	30.8 (20.1)	34.6 (21.7)	0.2685	36.3 (27.0)
Nausea/vomiting	8.2 (17.0)	12.6 (23.5)	0.1710	10.3 (19.7)
Pain	23.9 (24.1)	30.0 (23.6)	0.0487	30.9 (29.6)
Dyspnea	16.3 (22.9)	17.7 (22.4)	0.5714	20.4 (28.2)
Insomnia	29.0 (27.2)	31.7 (32.0)	0.7879	33.1 (32.6)
Appetite loss	19.2 (25.3)	20.2 (28.7)	0.9096	21.7 (31.0)
Constipation ^c^	16.3 (26.0)	18.9 (26.3)	0.6044	19.2 (28.8)
Diarrhea ^c^	8.7 (16.3)	12.8 (22.7)	0.3321	5.8 (15.2)
Total score	80.4 (13.8)	76.3 (16.7)	0.1172	NA

EORTC = European Organization for Research and Treatment of Cancer; QLQ-C30 = Quality of Life Questionnaire-Core 30; PRO = patient reported outcome; QoL = quality of lifer; SD = standard deviation; NA = Not available. ^a^ PRO-evaluable population is defined as all patients who have completed 1≥ PRO question at baseline and 1 ≥ PRO question after baseline. ^b^ Reference values for recurrent/metastatic breast cancer patients across all lines of treatment are shown. ^c^ Larger values better. ^d^ Larger values worse.

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
