# Peer review of "Patient-Reported Outcomes of Palbociclib Plus Exemestane with GnRH Agonist versus Capecitabine in Premenopausal Women with Hormone Receptor-Positive Metastatic Breast Cancer: A Prospective, Open-Label, Randomized Phase ll Trial (KCSG-BR 15-10)"

_cancers, 2020, doi:10.3390/cancers12113265_

Round 1

Reviewer 1 Report

This article reported the difference of PRO between HT and CT from prospective randomized P-II study. There was few similar reports, but we have already known the results which there were many difference in AE and Chemotherapy might lose to PRO.

I felt from these results that capecitabine was not invasive for patients. PRO was influenced from the tumor progression and treatment for AE. There was no details about them in the methods and discussion.

Did you schedule the analyze of estradiol change (Figure.5) for these study? You should indicate it in methods. If the authors did not indicate them in the protocol, these results should be indicated in discussion.

Author Response

1) I felt from these results that capecitabine was not invasive for patients. PRO was influenced from the tumor progression and treatment for AEs. There was no details about them in the methods and discussion.

--> You're right that PRO was influenced from the tumor progression and treatment for AEs. Based on the original study results, palbociclib plus ET showed better PFS than capecitabine. (20.1 months vs 14.4 month, HR=0.659, one-sided log-rank p=0.0235). That would be the reason why time to deterioration of patients with palbociclib plus ET had significant overall improvements and significant delay in many functioning and symptom scores. I mentioned this point in 3rd paragraph of the discussion part. 

--> In terms of capecitabine pivotal trial's data, there was no big difference of grade 3 AEs of capecitabine in this trial (file attached). Management of drug side effects was considered to be well performed at an average level.

2) Did you schedule the analyze of estradiol change (Figure.5) for these study? You should indicate it in methods. If the authors did not indicate them in the protocol, these results should be indicated in discussion.

--> Estradiol and FSH were measured based on the protocol. I added the hormone measurement as a 2.4 section. 

Reviewer 2 Report

The authors present their results from the work of patient-reported outcomes (PROs) on the clinical trial conducted by them utilizing palbociclib plus exemestane with GnRH agonist versus capecitabine in premenopausal patients with hormone receptor-positive metastatic breast cancer.  It is a very interesting study and I think that should be adopted if some minor points are revised.

  1. Although the authors analysed TTD of GHS/QoL, the minimally important difference (MID) of QoL should be also discussed.

  2. P2, L48: The Abbreviation of TTD should be specified.

  3. P9, L291: HR+HER MBC should be corrected to HR+HER2- MBC.

Author Response

  1. Although the authors analysed TTD of GHS/QoL, the minimally important difference (MID) of QoL should be also discussed.
    --> I specified this meaning of TTD of GHS/QoL in 2nd paragraph of discussion section.

  2. P2, L48: The Abbreviation of TTD should be specified.--> Corrected 

  3. P9, L291: HR+HER MBC should be corrected to HR+HER2- MBC.
    --> corrected 

Reviewer 3 Report

The manuscript submitted by Soohyeon Lee et al, which is well written and clearly presented, shows the patient-reported outcomes (EORTC QLQ-C30, quality of life questionnaire) from the YoungPearl study, a randomized open-label Phase 2 trial in premenopausal women with metastatic breast cancer (hormone receptor positive, and HER2 negative) to compare endocrine therapy (ET) plus palbociclib (CDK 4/6 inhibitor) with chemotherapy (capecitabine). Overall, in this study they show that quality of life was maintained upon therapy with ET + Palbociclib, similar to that found in patients treated with capecitabine.

Some minor points must be addressed:

  • Figure 2-1 and Figure 2-2 should be different figures, or just different panels within the same figure (e.g From A to M).
  • Please, show figures in the same order of the description of results. The authors describe Figure 2-1, and then Figure 3A and Figure 4A, and the backwards to Figure 2-2, and then forwards to Figure 3B and Figure 4 B-E. This system obliges the readers to jump from one page to others constantly, which is detrimental for the reading process.
  • Figures captions and axes, in Figures 2-1/2-2 and 4 are too small and difficult to read.
  • Including a list of abbreviations will help those readers who are not clinicians.

Author Response

  1. Figure 2-1 and Figure 2-2 should be different figures, or just different panels within the same figure (e.g From A to M). --> As your recommendation, I revised them within the same figure 2.
  2. Please, show figures in the same order of the description of results. The authors describe Figure 2-1, and then Figure 3A and Figure 4A, and the backwards to Figure 2-2, and then forwards to Figure 3B and Figure 4 B-E. This system obliges the readers to jump from one page to others constantly, which is detrimental for the reading process. --> You're right. In spite of them, I need to minimize the volume of the text (avoid some redundant description). I will ask to the editor this point to arrange the text and pictures so that readers can be viewed on one page.

    3. Figures captions and axes, in Figures 2-1/2-2 and 4 are too small and                    difficult to read. --> Texts enlarged 

  • Including a list of abbreviations will help those readers who are not clinicians. --> I'll ask to the editor whether abbreviations can be indicated for every figure on journal's style and then include them. 

Round 2

Reviewer 1 Report

The authors indicated the response to major points. I think that there are not so many new clinical information, but these data were very strict and correct. These data are useful for MBC patients and physicians. 

This manuscript is a resubmission of an earlier submission. The following is a list of the peer review reports and author responses from that submission.